# Graphical Time Warping for Joint Alignment of Multiple Curves

**Yizhi Wang**
Virginia Tech
yzwang@vt.edu

**David J. Miller**
Pennsylvania State University
djmiller@engr.psu.edu

**Kira Poskanzer**
University of California, San Francisco
Kira.Poskanzer@ucsf.edu

**Yue Wang**
Virginia Tech
yuewang@vt.edu

**Lin Tian**
University of California, Davis
lintian@ucdavis.edu

**Guoqiang Yu**
Virginia Tech
yug@vt.edu

## Abstract

Dynamic time warping (DTW) is a fundamental technique in time series analysis for comparing one curve to another using a flexible time-warping function. However, it was designed to compare a single pair of curves. In many applications, such as in metabolomics and image series analysis, alignment is simultaneously needed for multiple pairs. Because the underlying warping functions are often related, independent application of DTW to each pair is a sub-optimal solution. Yet, it is largely unknown how to efficiently conduct a joint alignment with all warping functions simultaneously considered, since any given warping function is constrained by the others and dynamic programming cannot be applied. In this paper, we show that the joint alignment problem can be transformed into a network flow problem and thus can be exactly and efficiently solved by the max flow algorithm, with a guarantee of global optimality. We name the proposed approach *graphical time warping* (GTW), emphasizing the graphical nature of the solution *and* that the dependency structure of the warping functions can be represented by a graph. Modifications of DTW, such as windowing and weighting, are readily derivable within GTW. We also discuss optimal tuning of parameters and hyperparameters in GTW. We illustrate the power of GTW using both synthetic data and a real case study of an astrocyte calcium movie.

## 1 Introduction

Time series, such as neural recordings, economic observations and biological imaging movies, are ubiquitous, containing rich information about the temporal patterns of physical quantities under certain conditions. Comparison of time series lies at the heart of many scientific questions. Due to the time distortions, direct comparison of time series using *e.g.* Euclidean distance is problematic. Dynamic time warping (DTW) is a powerful and popular technique for time series comparison using flexible warping functions. DTW has been successful for various tasks, including querying, classification, and clustering [1, 2, 3]. Although DTW is a mature approach, significant improvements have been proposed over the years, such as derivative DTW [4], weighted DTW [5], curve pairs with multiple dimensions [6], and extensions for large scale data mining [7].

However, DTW and all its variants consider the alignment of a *single* pair of time series, while in many applications we encounter the task of aligning multiple pairs simultaneously. One might apply DTW or its variants to each pair separately. However, very often, this is suboptimal because it ignores the dependency structure between the multiple warping functions. For example, when analyzing time lapse imaging data [8], we can consider the data as a collection of time series indexed by pixel. One

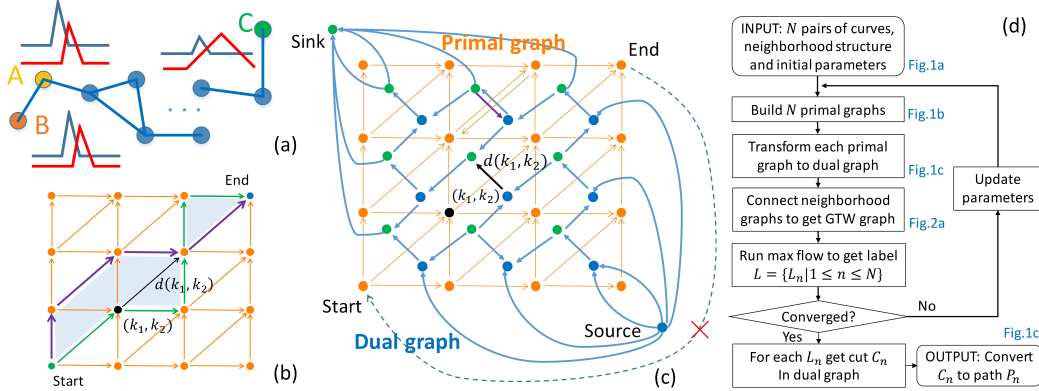

Figure 1: (a) Each node is a warping path between two curves $x_n$ and $y_n$. Neighboring paths are assumed to be similar (A and B) while non-neighboring ones may be quite different (A and C). (b) DTW can be represented as a shortest path problem in a directed graph. Each edge originating from node $(k_1, k_2)$ has a weight given by the dissimilarity (*e.g.* Euclidean distance) between $x_n(k_1)$ and $y_n(k_2)$. The *path* distance between the purple and green paths is defined as the area of the shaded parts. (c) Primal and dual graphs. The purple and gold edges are two infinite capacity reverse edges for the dual and primal graphs, respectively. Only two such edges are drawn for clarity. The dashed line shows the auxiliary edges used for transforming the primal graph to the dual graph, which are removed afterwards. (d) Flow chart for GTW. The corresponding figure for each step is annotated.

potential task is to compute the warping function associated with every pixel with respect to a given reference time series, with the ultimate goal of identifying signal propagation patterns among pixels. Although different pixels may have different warping functions, we expect that the functions are more similar between adjacent pixels than between distant pixels. That is, we expect a certain degree of smoothness among spatially adjacent warping functions. Another example is profile alignment for liquid chromatography-mass spectrometry (LC-MS) data, which is used to measure expression levels of small biomolecules such as proteins and metabolites. Each profile can be considered as a time series indexed by the retention time [9]. Typically, all profiles in the data set must be aligned to a reference profile. Because the LC-MS data measures similar things against a common reference profile, we expect similar warping functions for all profiles.

To the best of our knowledge, there is no existing approach that fundamentally generalizes DTW to jointly model multiple warping functions and align multiple curves, while retaining these advantageous properties of DTW: (1) computational efficiency and (2) a guarantee of global optimality. As we will discuss below, most existing efforts reuse DTW multiple times in a heuristic way. Interestingly, the necessity for and the challenge of a joint and integrated modeling approach come precisely from the two factors that contribute to the wide use of DTW. On one hand, the power of DTW is due to its flexibility in allowing a broad range of warping functions. As is well known in machine learning, an unavoidable consequence of flexibility is the problem of overfitting [10], and hence the estimated warping function is often unreliable. This problem becomes severe when the observed time series are very noisy and this is often the case, rather than the exception, for multiple curve alignment. On the other hand, the solution to DTW is extremely efficient and global optimality (with respect to the DTW objective function) is guaranteed, through the application of dynamic programming [11]. Unfortunately, when we consider joint modeling of multiple warping functions, dynamic programming is no longer applicable due to interactions between the different warping functions.

The computational burden of such a joint modeling seems prohibitive, and the feasibility of obtaining the global optimum is far from obvious, because each warping function is coupled to all the rest, either directly or indirectly. Thus, we were fortuitous to find that the joint modeling can be solved very efficiently, with global optimality ensured.

In this paper, we develop Graphical Time Warping (GTW) to jointly model multiple time warping functions in a unified framework. Given a set of warping function $\{P_n, n = 1, \ldots, N\}$ to be

optimized, a generic form of GTW can be expressed as follows:

$$\min_{\{P_n, n=1,...,N\}} \sum_{n=1}^{N} DTW\_cost(P_n) + \kappa \sum_{E(m,n) \in G_{struct}} dissimilarity\_cost(P_m, P_n), \quad (1)$$

where $P_n$ is subject to the same constraints as in conventional DTW such as boundary conditions, continuity, and monotonicity [12]. $G_{struct}$ is a graph encoding the dependency structure among the warping functions. Each node in the graph represents one warping function, indexed by $n$, and $E(m, n) \in G_{struct}$ denotes that there is an edge between nodes $m$ and $n$ in $G_{struct}$, whose corresponding warping functions are expected to be similar, as encoded in the second term of the cost (1). $DTW\_cost$ is the conventional DTW path finding cost and $dissimilarity\_cost$ ensures the neighboring warping functions are similar. The graph $G_{struct}$ can be defined by users or induced from other sources, which provides great flexibility for encoding various types of problems. For example, to analyze time series imaging data, the graph can be induced by the pixel grid so that edges exist only between spatially neighboring pixels. Alternatively, when aligning multiple LC-MS profiles, the graph is fully connected, such that each profile has an edge with all other profiles.

Since a warping function is a path in a two-dimensional grid from a given source to a given sink (as in Fig.1b), we propose to use the area bounded by two paths as the dissimilarity cost between them. Later, we will show how the optimization problem in Equation (1) equipped with this specific dissimilarity cost can be transformed into a network flow problem and solved by the max flow algorithm [13, 14].

As previously discussed, most DTW improvements have focused on the alignment of a single pair of curves. There are some heuristic efforts that deal with alignment of multiple curves. Chudova jointly performed clustering and time warping using a mixture model [15]; this assumes curves from the same cluster are generated by a single model. This is a suboptimal, restrictive "surrogate" for capturing the relationships between curves, and does not capture relationships as (user-)specified by a graph. Tsai et al. applied an MCMC strategy to align multiple LC-MS profiles with a single prior distribution imposed on all warping functions [9], but the approach is time-consuming and no finite-time convergence to the global optimum is guaranteed. Similarly, algorithms for aligning multiple DNA sequences are based on first clustering the sequences and then progressively aligning them [16, 17]. Most critically, all existing approaches assume special dependency structures, *e.g.* all nodes (curves) are equally dependent, and do not promise a globally optimal solution, while GTW works with any given dependency structure and finds the globally optimal solution efficiently.

Interestingly, the max flow algorithm has long been suggested as an alternative to DTW [13] by researchers in the network flow community. As an example, Uchida extended DTW to the non-Markovian case and solved it by the max flow algorithm [18]. Max flow formulations have also been developed to solve image segmentation [14], stereo matching [19] and shape matching problems [20]. But researchers in the time series analysis community have paid little attention to the max flow approach, perhaps because dynamic programming is much more efficient than the max flow algorithm and is sufficient for conventional DTW problems.

## 2  Problem Formulation

The task is to jointly align $N$ pairs of curves $(x_n, y_n), 1 \leq n \leq N$. For the sake of clarity, but without loss of generality, we assume all curves have the same length $K$ and each curve is indexed by an integer from 1 to $N$. To rigorously formulate the problem, we have the following definitions.

**Definition 1 – valid warping function**. A valid warping function for the $n_{th}$ pair of curves is a set of integer pairs $P_n = \{(k_{n,x}, k_{n,y})\}$ such that the following conditions are satisfied: (a) boundary conditions: $(1, 1) \in P_n$ and $(K, K) \in P_n$; (b) continuity and monotonicity conditions: if $(k_{n,x}, k_{n,y}) \in P_n$, then $(k_{n,x} - 1, k_{n,y}) \in P_n$ or $(k_{n,x}, k_{n,y} - 1) \in P_n$ or $(k_{n,x} - 1, k_{n,y} - 1) \in P_n$.

**Definition 2 – alignment cost**. For any given valid warping function $P_n$ and its corresponding pair of curves $(x_n, y_n)$, the associated alignment cost is defined as follows:

$$cost(P_n) = \sum_{(k_1, k_2) \in P_n} g(x_n[k_1] - y_n[k_2])), \quad (2)$$

where $g(\cdot)$ is any nonnegative function.

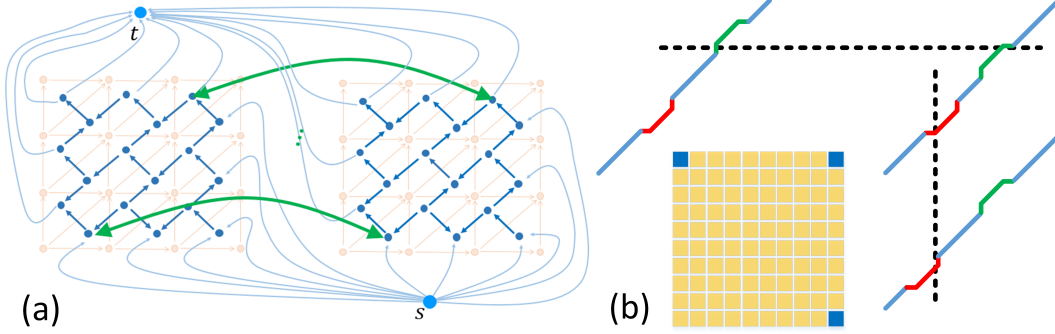

Figure 2: (a) GTW graph for two neighboring pairs. Only two (bidirectional) edges (green) are drawn for clarity. The orange background represents the (single pair) primal graphs. The blue foreground represents the dual graphs. (b) A neighborhood structure used for simulation. In the center is a 10 by 10 grid for 100 pairs, with *e.g.* a close spatial neighborhood defined around each grid point. The warping paths for the three blue squares are shown. The short red and green lines indicate when time shifts occur. They are at different positions along the three paths. The warping paths for spatially close pairs should be similar.

**Definition 3 – neighboring warping functions**. Suppose the dependency structure for a set of $N$ valid warping functions is given by the graph $G_{struct} = \{V_s, E_s\}$, where $V_s$ is the set of nodes, with each node corresponding to a warping function, and $E_s$ is the set of undirected edges between nodes. If there is an edge between the $m_{th}$ and $n_{th}$ nodes, we call $P_m$ and $P_n$ neighbors, denoted by $(m, n) \in Neib$.

**Definition 4 – distance between two valid warping functions**. We define the distance between two valid warping functions $dist(P_m, P_n)$ as the area of the region bounded by the two paths as shown in Fig.1b.

When we jointly align multiple pairs of curves, our goal is to minimize both the overall alignment cost and the distance between neighboring warping functions. Mathematically, denoting $V_P$ the set of valid warping function and $\kappa_1$ the hyperparameter, we want to solve the following optimization problem:

$$\min_P f(P) = \min_{P=\{P_n \in V_P | 1 \le n \le N\}} \sum_{n=1}^{N} cost(P_n) + \kappa_1 \sum_{(m,n) \in Neib} dist(P_m, P_n) \quad (3)$$

## 3 Methods

In this section, we first construct a graph based on Equation (3); then we prove that Equation (3) can be solved via the well-known max flow problem in the graph; finally we provide a practical algorithm.

### 3.1 Graph construction

**Definition 5 – directed planar graph for a single pair of curves**. For each pair of curves, consistent with the cost function (2), there is an induced directed planar graph [21], $G_n := \{V_n, E_n\}, 1 \le n \le N$, where $V_n$ and $E_n$ are the nodes and directed edges, respectively. An example is shown in Fig.1b.

**Definition 6 – dual graph**. Define $G'_n := \{V'_n, E'_n\}$ as the dual graph of the directed planar graph $G_n$, where nodes $V'_n$ are all faces of $G_n$, and for each $e \in E_n$, we have a new edge $e' \in E'_n$ connecting the faces from the right side of $e$ to the left side. This edge is directed (with positive direction by convention). The edge weights are the same as for the primal graph $G_n$. An example is shown in Fig.1c.

In contrast to conventional dual graph theory, one critical innovation here is that besides the positive edge we add in one more edge with reverse direction in the dual graph corresponding to each edge in

the primal graph. The weight for the reversed edge is set to infinity. This design is critical: otherwise, as demonstrated in Fig.3c, we cannot get an equivalent simpler problem.

**Definition 7 – GTW graph.** The GTW graph $G_{gtw} := \{V_{gtw}, E_{gtw}\}$ is defined as the *integrated graph* of all dual graphs $\{G'_n | 1 \leq n \leq N\}$ with the integration guided by the neighborhood of warping functions, such that $V_{gtw} = \{V'_n | 1 \leq n \leq N\}$ and $E_{gtw} = \{E'_n | 1 \leq n \leq N\} \cup \{(V'_{m,i}, V'_{n,i}) | (m, n) \in Neib\}$. All newly introduced edges $(V'_{m,i}, V'_{n,i})$ are bi-directional with capacity $\kappa_2$ (whereas all other edges have capacity proportional to the distance between two curves, measured at a pair of time points, *i.e.* $g(x_n(k_1) - y_n(k_2))$. An example is shown in Fig.2a.

## 3.2 Equivalent problem

We claim that the GTW problem as stated in Equation (3) is equivalent to the minimum cut problem on the GTW graph $G_{gtw}$ if we set $\kappa_2 = 2\kappa_1$.

## 3.3 Proof of equivalence

For brevity, more proofs of lemmas can be found in the supplementary material.

**Definition 8** – labeling of graph. $L$ is a *labeling* of graph $G$ if it assigns each node in $G$ a binary label. $L$ can induce a cut set $C = \{(i, j) | L(i) \neq L(j), (i, j) \in E_G\}$. The corresponding cut (or flow) is $cut(L) = cut(C) = \sum_{(i,j) \in C} weight(i, j)$, where $weight(i, j)$ is the weight on the edge between nodes $i$ and $j$.

Based on its construction, a labeling $L$ for the graph $G_{gtw}$ can be written as $L = \{L_n | 1 \leq n \leq N\}$, where $L_n$ is a labeling for the dual graph $G'_n$. So we can express the minimum cut problem for the graph $G_{gtw}$ as:

$$\min_{L} g(L) = \min_{L:=\{L_n | 1 \leq n \leq N\}} \sum_{n=1}^{N} cut(L_n) + \kappa_2 \sum_{(m,n) \in Neib} cut(L_m, L_n), \tag{4}$$

where $cut(L_n)$ is the cut of all edges for $G'_n$ and $cut(L_m, L_n)$ is the number of the cut edges between two neighboring dual graphs $G'_m$ and $G'_n$.

Denote $L_{mf}$ as the labeling induced by applying the max flow algorithm on $G_{gtw}$, where for each node $v$, $L_{mf}(v) = 0$ if $dist_{res}(v, s) < \infty$ and $L_{mf}(v) = 1$ if $dist_{res}(v, s) = \infty$, where $dist_{res}(i, j)$ is the distance between nodes $i$ and $j$ on the residual graph $G_{ext,res}$ given by the maximum flow algorithm and $s$ and $t$ are the source and sink nodes of $G_{gtw}$, respectively. Denote $S = \{v | L_{mf}(v) = 0\}$ and $T = \{v | L_{mf}(v) = 1\}$. We further denote $L_{mf,n}$ as the component corresponding to $G'_n$. Similarly, $S_n$ and $T_n$ are subsets of $S$ and $T$, respectively. Obviously, by the max-flow min-cut theorem, the resulting cut set $C_{mf}$ has the smallest cut. $C_{mf,n}$ is the cut set restricted to the graph $G'_n$.

**Lemma 1** Given labeling $L_{mf,n} \in L_{mf}$, $S_n$ forms a single connected area within graph $G'_n$. That is, $\forall i \in S_n$, there is a path with nodes $\{i, j, k, \ldots, s\} \subset S_n$ from $i$ to $s$. Similarly, $T_n$ also forms a single connected area. In other words, after applying the max flow algorithm on $G_{gtw}$, members of group $S_n$ do not completely surround members of group $T_n$, or vice versa.

**Definition 9** – directed cut set. Cut set $C$ is a *directed cut set* if $\forall (i, j) \in C$, either $i \in S$ and $j \in T$ or $cap(i, j) = \infty$, $i \in T$ and $j \in S$. As will be seen, this definition ensures that the cut set includes only the edges $e'$ that correspond to edges in the primal graph $G_n$, instead of the reverse edges introduced when building the dual graph $G'_n$, which give the wrong path direction.

**Lemma 2** $C_{mf,n}$ is a directed cut set.

From Lemma 1 and 2, we can build the link between the first term of $f$ (Equation (3)) and $g$ (Equation (4)).

**Lemma 3** For any directed cut set $C_n, 1 \leq n \leq N$ for $G_{gtw}$, there is a valid warping function $P_n, 1 \leq n \leq N$ for $G_n, 1 \leq n \leq N$ so that $cut(C_n) = cost(P_n)$, and vice versa.

**Lemma 4** For two neighboring pairs $(x_m, y_m)$ and $(x_n, y_n)$, $dist(P_n, P_m) = 0.5|C_{m,n}|$, where we denote $C_{m,n} := \{(V'_{m,i}, V'_{n,i}) | V'_{m,i} \in S, V'_{n,i} \in T \text{ or } V'_{m,i} \in T, V'_{n,i} \in S\}$.

Lemma 4 states that the distance between two paths in the primal graph (Fig.1b) is the same as the number of neighborhood cuts between those two pairs, up to a constant scaling factor.

**Lemma 5** Let $P$ be a set of valid warping functions for $\{G_n|1 \leq n \leq N\}$ and let $L$ be the labeling in $G_{gtw}$ that corresponds to directed cuts. If $\kappa_2 = 2\kappa_1$, given $P$, we can find a corresponding $L$ with $f(P) = g(L)$ and given $L$, we can find a corresponding $P$ so that $g(L) = f(P)$.

*Proof.* First we show each $P$ gives an $L$. As each path $P_n$ can be transformed to a directed cut $C_n$ (Lemma 3), which by definition is also a cut, it gives a valid labeling $L_n$ and $cost(P_n) = cut(L_n)$ by definition. And $dist(P_m, P_n) = 0.5 \times cut(L_m, L_n)$ by Lemma 4. Then, with $\kappa_2 = 2\kappa_1$, we find $L = \{L_n|1 \leq n \leq N\}$ such that $f(P) = g(L)$. Conversely, given $L = \{L_n|1 \leq n \leq N\}$ corresponding to directed cuts $C_n$, $C_n$ can be transformed back to a valid path $P_n$ with the same cost (Lemma 3). For the cut between $L_m$ and $L_n$, we still have $cut(L_m, L_n) = 2 \times dist(P_m, P_n)$ using Lemma 4. Thus we find a set $P = \{P_n|1 \leq n \leq N\}$ with the same cost as $L$.

**Theorem 1** If $L_{mf}$ is a labeling induced by the maximum flow algorithm on $G_{gtw}$, then the corresponding $P$ minimizes $f(P)$.

*Proof.* Assume the max flow algorithm gives us a labeling $L$, which corresponds to path $P$ and by Lemma 5 the relationship $f(P) = g(L)$ holds. Here $f$ is the primal cost function and $g$ is the dual cost function. Assume we have another labeling $L' \neq L$ and it corresponds to another path $P'$; then also by Lemma 5 $f(P') = g(L')$ holds. Suppose path $P'$ is better than path $P$, i.e. $f(P') < f(P)$. This implies $g(L') < g(L)$, which contradicts the assumption that $L$ is the labeling from the max flow algorithm. Thus, there is no better path in terms of $f()$ than that associated with the result of the max flow algorithm.

From Theorem 1 we know that after the max flow algorithm and labeling finishes, we can get a single path $P_n$ for each pair $(x_n, y_n)$, which solves the primal form optimization problem. Since the labeling sometimes may not be unique, different labelings may have the same cut. Correspondingly, different paths in the primal graph may have the same (jointly minimum) cost.

**Corollary 1** If $\kappa_1 = \kappa_2 = 0$, $L$ that minimizes $g(L)$ corresponds to the $P = \{P_n|1 \leq n \leq N\}$ where $P_n$ is the solution of the single pair DTW problem for $(x_n, y_n)$.

### 3.4 Flowchart of GTW algorithm

Once the equivalence is established, a practical algorithm is readily available, as shown in the flowchart of Fig.1d. Assuming the hyperparameter ($\kappa_1$) is fixed, one first constructs a primal graph separately for each alignment task, then converts each primal graph to its dual form, and finally adds in edges to the set of dual graphs to obtain the GTW graph. Once we get the GTW graph, we can apply any maximum flow algorithm to the graph, leading to the minimum cut set $C_{mf}$. For each sub cut-set $C_{mf,n}$ corresponding to the $n_{th}$ dual graph $G'_n$, we convert the cut edges back to edges in the primal graph $G_n$. The resulting edges will be connected as a warping path and hence lead to a warping function. The set of resultant warping functions are the solution to our GTW problem. A working example is given in the Supplementary.

Note also that, as indicated in Fig.1d, this algorithm can be iteratively applied, with parameter (*and hyperparameter*) re-estimation performed at each iteration. The primary parameter is the noise variance (which can easily be generalized to a separate noise variance parameter for each pair of curves, when appropriate). In addition to the major hyperparameter $\kappa_1$ in Equation (3), we may use other hyperparameters to incorporate prior knowledge such as favoring a diagonal warping direction, which actually results in an extension of DTW even for a single pair of curves. In the Supplement, we show that the hyperparameters can be tuned, along with parameters, via either cross validation or approximately consistent with maximum likelihood estimation. In addition, as a heuristic rule of thumb, we can choose $\kappa_1 = a\sigma^2$, where $\sigma^2$ is the noise variance and $a \in (1, 10)$.

## 4  Experimental results

We used synthetic and real data to compare the performance of GTW and DTW. For the synthetic data, we evaluate the performance by the estimation error for the warping path $P_n$. For real data, we examine the spatial delay pattern relative to a reference curve. We also illustrate the impact of the capacity of the reverse edges. More experiments can be found in the Supplement.

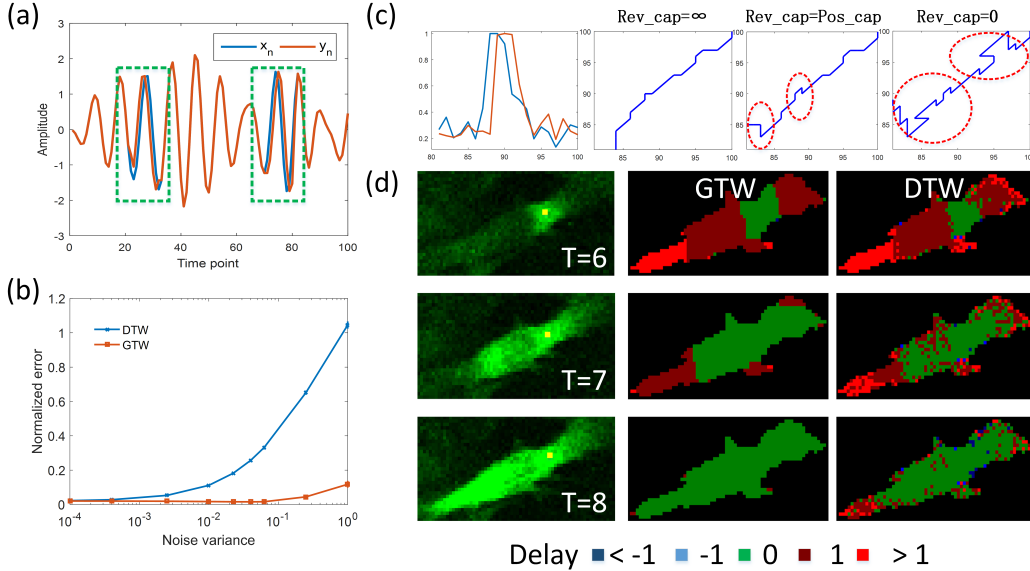

Figure 3: (a) The curves before (blue, $x_n$) and after (red, $y_n$) warping in the simulation. The green dashed squares indicate where the warping occurs. (b) Performance comparison of GTW and DTW for 100 simulations under different additive noise variances. Both cases include the off-diagonal weights $\beta$ (see section 4.1). Error bars indicate standard deviation. (c) The impact of reverse capacity. Left: a pair of curves from an astrocyte imaging movie. Only times 81 to 100 are shown. The right three figures are the warping paths with different reverse capacities. $Pos\_cap$ is the capacity for corresponding edges from the primal graph. Red dashed circles indicate where the DTW constraints are violated. (d) Estimated propagation patterns on the astrocyte image. Left: original movie from times 6 to 8. The yellow dot is the position of the reference curve. Middle: the delay pattern of pixels relative to the reference curve, estimated by GTW. Right: results for DTW.

## 4.1 Experiment on synthetic data

We generated $N = 100$ pairs of curves $(x_n, y_n)$. Each pair is linked by a warping function $W_n$ so that $y_n = W_n(x_n)$. Curve $x_n$ is a time series composed of low pass filtered Gaussian noise and $y_n$ is generated by applying $W_n$ on $x_n$ (Fig.3a). Noise is also added to both $x_n$ and $y_n$. In this simulation the pairs are in a $10 \times 10$ four connected grid; thus the ground-truth warping paths for neighboring pairs are similar (Fig.2b). The warping path of the pair at location $(1, 1)$ has a one time-point shift from 21 to 30 and another one from 71 to 80. The pair at location $(10, 10)$ has a one time point shift from 30 to 39 and another from 62 to 71. The warping function for pairs between these locations are smoothly interpolated.

We ran the simulation 100 times and added uncorrelated Gaussian noises to $x_n$ and $y_n$. All hyperparameters were initialized to 0; the noise variance was initialized to 0.01. In addition, the distance of the path from the diagonal line was penalized via a hyperparameter $\beta = \sqrt{d}/\sigma^2$, where $d$ is the distance of a point in the path to the diagonal. When the parameter and hyperparameter changes were all less than 0.001, we stopped the algorithm. Convergence usually occurred within 10 iterations. The estimated path was compared with the ground truth one and we define the normalized error as

$$err_{norm} = \frac{1}{(K-1)N} \sum_{k=1}^{K-1} \sum_{n=1}^{N} \left| S_{\hat{P}_n}(k, k+1) - S_{P_n}(k, k+1) \right| \qquad (5)$$

Here $S_{\hat{P}_n}(k, k+1)$ is the area under the path $\hat{P}_n$ between times $k$ and $k+1$.

**GTW improves the accuracy in estimating warping functions**. As shown in Fig.3b, GTW outperforms DTW even when the noise level is small or moderate. Moreover, while DTW degrades with increasing noise, GTW maintains a much smaller change in its normalized error for increasing noise.

**Infinite capacity reverse edges are critical**. In Fig.3c we illustrate the importance of introducing infinite capacity reverse edges when we construct the dual graph $G'_n$ for each primal graph $G_n$. This ensures the cut found by the maximum flow algorithm is a directed cut, which is linked to a path in the primal graph that satisfies the constraints of DTW. If the reverse edge is not added, the max flow algorithm acts as if there is a reverse edge with zero weight. Alternatively, we can add in a reverse edge with the same weight as for the positive direction. However, in both cases as shown in the right two subplots of Fig.3c, DTW's monotonicity and continuity constraints are violated almost everywhere, since what we obtain by max flow in this case is no longer a directed cut and the path in the primal graph is no longer a valid warping function.

## 4.2 Application to time-lapse astrocyte calcium imaging data

We applied GTW to estimate the propagation patterns of astrocyte calcium fluorescent imaging data [22, 8]. The movie was obtained from a neuro-astrocyte co-cultured Down syndrome cell line. It contains 100 time points and rich types of propagation are observed during the time course. Here we focused on a selected region. The movie between time instants 6 and 8 is shown in the left column of Fig.3d. At time 6, the activity occurs at the center part and it spreads out over the subsequent time points. At time 8, the active area is the largest. Since the movie was taken while the cells were under drug treatment conditions, the properties of these calcium waves are important features of interest. Here we focused on one segmented area and identified the propagation pattern within it. We extracted the curve for one pixel as the reference curve $x$ (Fig.3c, left) and all other pixels are $y_n$. So now $x_1 = x_2 = \cdots = x_N = x$, which is a special case of GTW. All parameters and hyperparameters were initialized in the same way as previously and both methods included an off-diagonal cost $\beta$. From the estimated warping path, we extracted the delay relative to the reference curve, which is defined as the largest discrepancy from the diagonal line at a given time point (Fig.3d, middle and right columns). GTW gives cleaner patterns of delay compared to DTW, which produces noisier results.

## 5 Discussion

While GTW can be applied to time series data analysis tasks like classification and clustering to obtain a smoothed distance measure, it could be even more powerful for mining the relationships between warping functions. Their differences could be classified or clustered, and explained by other features (or factors) for those curve pairs. This may bring further insights and interpretability to the solution. As a two-layer network for time series, GTW is a general framework for analyzing the pattern of warping functions. First, the time series can be flexibly organized into pairs with DTW constraints. One curve can participate in multiple pairings and even play different roles (either as a reference or as a test curve). Partial matching, direction preference and weighting of DTW can be readily incorporated. In addition, GTW allows the test curve and the reference curve to have different lengths. Second, the construction of graphs from pairs adds another layer of flexibility. For spatio-temporal data or video analysis, physical locations or pixels naturally guide the choice of graph edges. Otherwise, we can avoid using a fully connected graph by utilizing any auxiliary information on each pair of curves to build the graph. For example, features related to each subject (*e.g.*, clinical features) can be used to enforce a sparse graph structure.

## 6 Conclusion

In this paper, we developed graphical time warping (GTW) to impose a flexible dependency structure among warping functions to jointly align multiple pairs of curves. After formulating the original cost function, the single pair time warping term is transformed into its dual form and pairwise costs are added. We proved the equivalence of this dual form and the primal form by the properties of the dual-directed graph as well as the specific structure of the primal single pair shortest path graph. Windowing, partial matching, direction, and off-diagonal costs can also be incorporated in the model, which makes GTW flexible for various applications of time warping. Iterative unsupervised parameter estimation and inference by max flow are shown to be effective and efficient in our experiments. Simulation results and a case study of astrocyte propagation demonstrate the effectiveness of our approach.

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
