[Supplementary Material · Graphical Time Warping for Joint Alignment of Multiple Curves - Supplement.pdf]

# Supplementary Material for "Graphical Time Warping for Joint Alignment of Multiple Curves"

## CONTENTS

## 1 LIST OF SYMBOLS

| Symbol | Description |
|---|---|
| $P_n$ | Path in DTW for single pair $(x_n, y_n)$ |
| $W_n$ | Warping function for $n_{th}$ pair $(x_n, y_n)$ |
| $K$ | Number of time points in each time series or curve |
| $k_1, k_2$ | Index of element in a sequence |
| $L$ | Labels for either single or multiple pair dual graphs |
| $L(i)$ | Label for the $i_{th}$ node |
| $N$ | Total number of pairs |
| $i, j$ | Node ID in dual graph |
| $(i, j)$ | Directed edge in dual graph from $i$ to $j$ |
| $m, n$ | Indices for pairs of curves |
| $s, t$ | Source and sink in dual graph $G'_n$ or $G_{gtw}$ |
| $S, T$ | Node groups corresponding to $s$ and $t$ |
| $C, C_n$ | Sets of edges in the cuts for GTW and a dual graph $G_n$ |

| | |
|---|---|
| $G_n, G'_n, G_{gtw}$ | Graph for the $n_{th}$ pair, dual for a pair, GTW graph |
| $E_n, E'_n, E_{gtw}$ | Edges for the above graph. $E'_n$ may also mean a subset of $E_{gtw}$ |
| $V_n, V'_n, V_{gtw}$ | Nodes for the above graph. $V'_n$ may also mean a subset of $V_{gtw}$ |
| $C_{mf}, L_{mf}$ | Cut and labels obtained from maximum flow algorithm |
| $V_{n,i}$ | The $i_{th}$ element of the ordered node set $V_n$ |
| $e, e'$ | Edges in primal and dual graphs, respectively |
| $f(P), g(L)$ | Primal and dual form cost functions |
| $\phi$ | Empty set |
| $|X|$ | Cardinality of set $X$ |
| $\kappa$ | Smoothness cost coefficient |
| $\alpha$ | Direction cost coefficient |
| $\beta$ | Off-diagonal distance cost coefficient |
| $\sigma^2$ | Noise variance |
| $cost(P_n)$ | The cost for a single warping path |
| $dist(P_m, P_n)$ | Distance between warping paths for pair $m$ and $n$ |
| $cap(e), cap(i,j)$ | Capacity of the directed edge $e$ from $i$ to $j$ in the dual graph |
| $weight(e)$ | Weight for edge $e$ in primal graph $G$ |
| $cut(L), cut(C)$ | Cut for labeling $L$ or corresponding cut set $C$, which is the sum of capacities of involved edges from $S$ to $T$ |
| $U(i,j), V(i,j)$ | Data cost and smoothness cost |

## 2    A WORKING EXAMPLE OF GTW

In this section we provide a step by step walk through for graphical time warping, starting from the raw data. Assume we get 5 time-series, or curves, each with length 4:

$$x_1 = [0.0511 \quad 1.0036 \quad 0.0103 \quad 0.1555]$$

$$x_2 = [0.0426 \quad 0.9392 \quad 0.8486 \quad -0.0145]$$

$$x_3 = [0.0362 \quad 0.9849 \quad 1.0829 \quad -0.0198]$$

$$x_4 = [0.1004 \quad 0.0832 \quad 0.9972 \quad 0.0272]$$

$$x_5 = [-0.1651 \quad -0.1167 \quad 1.2430 \quad -0.0521]$$

The curves are shown in Supplement Figure 1 (left). Curve $x_2$ looks similar to $x_3$ while $x_4$ is similar to $x_5$. $x_1$ is different from the remaining curves.

### 2.1    Determining curve pairings

Given the data, the first thing to do is to determine the pairings between curves. Assume here we know that curves $x_2$ through $x_5$ are generated from curve $x_1$; then the pairings could be: $(x_1, x_2)$, $(x_1, x_3)$, $(x_1, x_4)$, and $(x_1, x_5)$. The order of elements in a pairing is important since the warping path from $x_1$ to $x_2$ is usually different from the warping path from $x_2$ to $x_1$.

### 2.2    Determining graph structure

After pairing the curves, we treat each pair as a node and need to determine the structure of the graph that consists of those nodes. For example, if we know curve $x_2$ is related to $x_3$, $x_3$ is related to $x_4$, $x_4$ to $x_5$, and $x_5$ to $x_2$, we might reasonably suppose that after all of them are paired with $x_1$, the pairs preserve

these relationships. The graph is shown in Supplement Figure 1 (right). Note that, different from a pairing between curves, the relationships defined between *pairs* are un-directed. After applying GTW, we will get the warping functions that map $x_1$ to $x_2$, $x_1$ to $x_3$, $x_1$ to $x_4$, and $x_1$ to $x_5$, under the constraints of the graph.

**Supplement Figure 1. Left: five observed curves. Right: graph structure between pairs of curves. Each node is a curve pair. The relationship between pairs is not directional, so we use edges without arrows.**

## 2.3 Building dual graphs for each pair

For each pair of curves, we need to build primal and dual graphs. We use pair $(x_1, x_2)$ as an example. Graphs for all the other pairs can be constructed in exactly the same way. We first compute the $4 \times 4$ distance matrix $D_{12}$:

$$D_{12} = \begin{bmatrix} 0.0001 & 0.7886 & 0.6360 & 0.0043 \\ 0.9235 & 0.0042 & 0.0240 & 1.0365 \\ 0.0010 & 0.8627 & 0.7027 & 0.0006 \\ 0.0127 & 0.6141 & 0.4803 & 0.0289 \end{bmatrix}$$

Each element of $D$ is the Euclidean distance between a sample from $x_1$ at one time instant and a sample from $x_2$ at one time instant. For example, the element in the second row and third column (shown as red) is the cost of mapping the second element of $x_1$ to the third element of $x_2$. In other words, $D_{12}(2,3) = [x_1(2) - x_2(3)]^2$. For the other three pairs, we compute in the same way the distance matrices $D_{13}$, $D_{14}$, and $D_{15}$.

Next we build the dual graphs for each pair, assigning a weight to every edge of these graphs. Let us continue using pair $(x_1, x_2)$ as an example. For simplicity, we assume the first time point of $x_1$ is aligned to the first time point of $x_2$ and the last time point of $x_1$ is aligned to the last time point of $x_2$. The primal graph, along with the edge weights, are shown in Supplement Figure 2. In the implementation, we only need to build the dual graph. We show the primal graph here to make the construction of the dual graph easier to understand.

Now we discuss the details in building the primal graph. As there are four time points in $x_1$ and four time points in $x_2$, we build a graph with $4 \times 4 = 16$ nodes. Each node represents aligning one time point of $x_1$ to another time point in $x_2$. We connect those nodes in the way shown in Supplement Figure 2 (left). The bottom left green node represents aligning the first time point of $x_1$ to the first time point of $x_2$ while the top right blue node represents aligning the fourth time point of $x_1$ to the fourth time point of $x_2$. Then we can assign weights to each edge. Let's consider node $(2,2)$, drawn with a purple dot, which means aligning the second time point of $x_1$ to the second time point of $x_2$. There are three edges originating from that

one and we draw them as purple lines with arrows. We might assign the weights of all three edges as $D_{12}(2,2)$, where $D_{12}$ is the distance matrix for pair $(x_1, x_2)$.

**Supplement Figure 2.** Primal graph for pair $(x_1, x_2)$. The sub-figures on the bottom and left of each graph are the two curves in that pair. Left: the weight of each edge corresponds to an element in distance matrix $D_{12}$. Right: the weight for each edge is shown in the graph.

**Supplement Figure 3.** Dual graph for pair $(x_1, x_2)$. The reverse edges with infinite capacity are not shown.

Of course, we can assign different weights for those three edges. For example, we can assign lower weight for the edge along the diagonal. In this way, we prefer directions of the warping path that change less frequently. This is one advantage of assigning weights to edges, instead of nodes, though the main reason is to allow the dual graph formulation that allows the constraints between curve pairs.

For some nodes, there are less than three edges that originate from them. For example, node $(4,1)$ has only one edge starting from it and the weight for it is $D_{12}(4,1)$. For the top right node, there is no edge starting from it. This is easy to understand since as we require the last time point of $x_1$ must align to the last time point of $x_2$, the cost of aligning those two time points has no impact on the warping path – this node must be on the warping path. The weights for all edges in the primal graph for pair $(x_1, x_2)$ are shown in Supplement Figure 2 (right).

The dual graph can be immediately obtained once we have the primal graph, as shown in Supplement Figure 3. Each face in the primal graph is a node in the dual graph. If, in the primal graph, two faces are separated by an edge, then this edge corresponds to an edge in the dual graph that connects the two nodes that originate from these two faces. The direction of the edge in the dual graph is obtained by a counterclockwise rotation of the corresponding edge in the primal graph. The capacity of each dual edge is the weight for the corresponding edge in the primal graph. For each dual edge, we need to add another edge with opposite direction, whose weight is infinite. Those edges are not shown in Supplement Figure 3, but must be specified in the implementation for most max flow algorithm software packages.

**Supplement Figure 4. Complete graph with four pairs. Only several edges are drawn between pairs for clarity, though all corresponding nodes between pairs specified by** Error! Reference source not found. **need to be connected (except for nodes $s$ and $t$). Note that there is only one node $t$ and one node $s$. They are shown in the dual graph for each pair for clarity.**

## 2.4 Connecting the single pair dual graphs to the GTW graph

As we already have four dual graphs, each for one pair, we can now connect those graphs according to the graph structures shown in Supplement Figure 1 (right). First let's consider the edge in Supplement Figure 1 (right) that connects pair $(x_1, x_2)$ and pair $(x_1, x_3)$. To connect those two pairs, we simply add an un-directed edge between each pair of corresponding nodes in the two dual graphs, as shown in Supplement Figure 4. Similarly, we repeat this for all other edges required by Supplement Figure 1 (right).

These newly introduced edges between pairs do not have directions. However, as the edges in each dual graph have directions, we need to replace these un-directed edges with directed ones, unless the max flow routine can support a mix of directed and un-directed edges. This can be simply done by replacing the un-directed edge $A \leftrightarrow B$ by two directed edges $A \rightarrow B$ and $B \rightarrow A$ that have the same capacity as $A \leftrightarrow B$.

**Supplement figure 5. Left: GTW gives labelling and cutting edges for nodes on the dual graph for pair $(x_1, x_2)$. We do not need to consider edges across pairs. Nodes are labelled as either 0 (blue nodes, source, $s$) or 1 (green nodes, sink, $t$). The cutting edges are drawn in thick blue arrow. Right: from cutting edges we can recover the warping path. The path is drawn in thick orange arrows.**

**Supplement Figure 6. Warping paths from GTW. All four curve pairs are also shown.**

The capacity for each edge we introduced here is the smoothness parameter $\kappa$, which can usually be chosen by cross validation or maximum likelihood-based methods. Here we simply choose $\kappa = 2\sigma^2 = 0.02$ for all edges introduced here, though it is possible to choose different values for different edges.

Finally, note that although in Supplement Figure 4 node $s$ and node $t$ are drawn in the dual graph for each pair, there must be just one node $s$ and one node $t$ in the implementation. In other words, all $s$ and $t$ shown in Supplement Figure 4 must be treated as the same node, respectively.

## 2.5 Solving the problem and obtaining the path

Given the graph shown in Supplement Figure 4, we can apply the max flow algorithm to label each node as either 0 or 1, which correspond to source and sink, respectively. Then our task is to recover the path for each pair from the labeling. We still use pair $(x_1, x_2)$ as an example. The labelling for the dual graph for pair $(x_1, x_2)$ is shown in Supplement figure 5 (left), where the green dots are labelled as 1 and blue dots are labelled as 0. We do not need to consider the edges across single pair dual graphs. Usually the max flow algorithm can tell you where the cut happens. Those cutting edges are drawn in thick dark blue lines. Given those cutting edges in dual graph, we need to find the corresponding edges in the primal graph, which are draw as thick orange lines in Supplement figure 5 (right). Connecting those orange edges, we obtain the warping path for pair $(x_1, x_2)$. Repeat this process, we can get the paths for all pairs, as shown in Supplement Figure 6.

## 3 PROOFS

**Lemma 1** Given labeling $L_{mf,n} \subset L_{mf}$, $S_n$ forms a single connected area within graph $G_n'$. That is, $\forall i \in S_n$, there is a path with nodes $\{i, j, k, \dots, s\} \subset S_n$ from $i$ to $s$. Similarly, $T$ also forms a single connected area.

*Proof*. Assume node set $T_0 \subset T$ is surrounded by $S$. Then for any node $t_0 \in T_0$, we can always construct a path from $t$ to $t_0$ along the infinite capacity reverse edges (Supplementary Fig.1, left). But this path must include an adjacent pair $s_1 \in S$ and $t_1 \in T_0$ at some place, since $T_0$ is completely surrounded by $S$. Then it cannot be a minimum cut -- by relabeling all nodes in $T_0$ to $S$, we will have an even smaller cut. But the max flow algorithm should have already given the min cut. Similarly, assume $S_0$ is surrounded by $T$. For any node $s_0 \in T$, we can always construct a path from $s_0$ back to $s$ along the reverse edges. But this path must include an adjacent pair $s_2 \in S_0$ and $t_2 \in T$ at some place since $S_0$ is completely surrounded by $T$. Then it cannot be a minimum cut either. □

Lemma 1 ensures that a valid labeling for $G_{gtw}$ is also valid for $G_n$, which means all nodes in $S$ are reachable by $s$.

**Lemma 2** $C_{mf,n}$ is a directed cut within $G_n'$ given $L_{mf,n} \subset L_{mf}$.

*Proof*: Due to reverse edges, the $C_{mf}$ that contains a reverse edge $(i, j)$ so that $i \in S$ and $j \in T$ will give infinite cut. So $C_{mf}$ is a directed cut. Since $C_{mf,n} \subset C_{mf}$, it is also a directed cut. □

**Lemma 3**. For directed cut $C_n$ for $G_{gtw}$, there is a set of valid warping paths $\{P_n | 1 \le n \le N\}$ for $\{G_n | 1 \le n \le N\}$ so that $cut(C_n) = cost(P_n)$, and vice versa.

*Proof*. Since $C_n$ is a directed cut for subgraph $G_n'$ (Lemma 2), for edge $e' \in C_n$ in $G_n'$, there is a corresponding primal directed edge $e \in E_n$ according to the construction of $G_n'$ from $G_n$ (Definition of the dual graph). Since $C_n$ is a directed cut, the collection $P_n^{aux} := P_n \cup \{e_0\} = \{e' | e' \in C_{mf,n}\} \cup \{e_0\}$ forms a

circuit in $G_n$ since each directed cut in the dual graph is a circuit in the primal graph (corollary 2.44 in [1]). Here $e_0$ is the auxiliary edge (Fig.1c), which corresponds to auxiliary cut $(t,s)$. Removing $e_0$, $P_n$ is a path from source to sink in the primal graph, where the reverse edges do not exist. Then we show $P_n$ satisfies all constraints of DTW.

**Supplement Figure 7. One node whose label is completely surrounded by another group. By finding paths to s or from t, we show this case cannot occur as part of a min cut. Left: T_0 surrounded by S. Right: S_0 surrounded by T.**

*Boundary conditions*: since $P_n^{aux}$ is a loop that must go through the auxiliary edge from sink to source, $P_n$ must go through source and sink, which is the desired boundary point to start and stop.

*Monotonicity*: all edges defined in the primal graph $G_n$ go from time matching pair $(k_1, k_2)$ to $(k_1', k_2')$ so that $k_1 \leq k_1'$ and $k_2 \leq k_2'$ but $(k_1, k_2) \neq (k_1', k_2')$. So all edges in $P_n$ satisfy this.

*Continuity*: in the version of DTW we considered here, edges can only connect $(k_1, k_2)$ with $(k_1 + 1, k_2)$, $(k_1, k_2 + 1)$ or $(k_1 + 1, k_2 + 1)$, so the continuity constraints are also satisfied by $P_n$.

Since for each corresponding edge $e$ and $e'$, $cap(e') = weight(e)$, we have

$$cut(C_n) = \sum_{e' \in C_n} cap(e') = \sum_{e \in P_n} weight(e) = cost(P_n).$$

Conversely, for each $P_n$, we transform every $e \in P_n$ to $e'$ with the auxiliary edges introduced as above. Then this loop will become a directed cut $C_n = \{e'\}$ in the dual. □

**Lemma 4** For two neighboring pairs $(x_m, y_m)$ and $(x_n, y_n)$, if $L_m$ and $L_n$ correspond to directed cuts, $dist(L_n, L_m) = 0.5|C_{m,n}|$, where we denote $C_{m,n} := \{(V'_{m,i}, V'_{n,i})|V'_{m,i} \in S, V'_{n,i} \in T \text{ or } V'_{m,i} \in T, V'_{n,i} \in S\}$.

*Proof.* As $L_m$ corresponds to directed cut $C_m$, by Lemma 3, we get a valid warping path $P_m$ from $C_m$. Similarly, we get a valid $P_n$ from $L_n$. Assume we put time points of $x$ on the $x$ axis. Consider time segment $\Delta_k = [k, k+1]$. In $G_m$, assume during $\Delta_k$, $e_m = \{(k, k_m), (k+1, k_m)\} \in P_m$. Then the area under $\Delta$ ($S_{\Delta_k m}$) is $k_m$. Similarly, if, during $\Delta_k$, $P_n$ goes from $e_n = \{(k, k_n), (k+1, k_n)\} \in P_n$, $S_{\Delta_k n} = k_n$. The difference of areas is $|k_m - k_n|$.

For $(x_m, y_m)$, during $\Delta_k$, $e_m$ gives an area $A_m$ bounded by $(k, 1)$, $(k+1, 1)$, $(k, k_m)$ and $(k+1, k_m)$ in $G_n$. Faces within it become nodes $V$ in $G'_n$ and $V \subset S$. Otherwise, assume $v \in V'_m$ and $v \in T$. Utilizing the structure of $G'_m$, we can always find $v' \in V'_m$ within $A_m$ so that $(v, v') \in E'_m$ and $v' \in S$.

This will induce a cut with infinite cost. We have $|V| = 2k_m$. Similarly, for pair $(x_n, y_n)$, the above edge gives $|V| = 2k_n$. So the number of cuts will be $2|k_m - k_n|$.

So for duration $\Delta_k$ with the above edge directions for both pairs, $dist(L_n, L_m)_{\Delta_k} = 0.5|C_{m,n,\Delta_k}|$, where $C_{m,n,\Delta_k}$ is the neighborhood cuts within $\Delta_k$. The proofs for other direction combinations are the same as above. So we have

$$dist(L_n, L_m) = \sum_{k=1}^{K} dist(L_n, L_m)_{\Delta_k} = \sum_{k=1}^{K} 0.5|C_{m,n,\Delta_k}| = 0.5|C_{m,n}|. \qquad \square$$

**Corollary 1**. If $\kappa_1 = \kappa_2 = 0$, $L$ that minimizes $g(L)$ corresponds to the $P = \{P_n | 1 \leq n \leq N\}$ where $P_n$ is the solution of the single pair DTW for $(x_n, y_n)$.

*Proof.* From Theorem 1, using the maximum flow results $L$ we can get $P$ that minimizes $f(P)$ as $\kappa_1 = 0.5\kappa_2$ can still be applied. Then

$$\max_{P=\{P_n|1\leq n\leq N\}} f(P) = \max_{P=\{P_n|1\leq n\leq N\}} \sum_{n=1}^{N} cost(P_n) = \sum_{n=1}^{N} \max_{P_n} cost(P_n).$$

So $P_n$ solves the single pair DTW problem. $\qquad \square$

## 4  EXTENSIONS TO SINGLE PAIR DTW

The original form of DTW needs to be extended in practice to account for real applications. First, windowing is widely used to reduce the computational cost and avoid spurious warping [2]. Second, since the first/last time point in $x_n$ does not necessarily correspond to the first/last point in $y_n$, we need to allow partial matching. Third, for different data sets, different path characteristics may be most realistic, *e.g.* how rapidly the direction should change [3]. Finally, paths that deviate far from the diagonal may need to be penalized [4]. The former two are achieved by modifying the structure of the graph and the latter two by changing the cost function. Our established theoretical results also hold under these modifications of DTW. The dual graph approach in GTW requires the original DTW graph to be planar. However, as discussed in [2] other step patterns that cannot be represented by planar graphs make little difference empirically from the classical pattern GTW used in this paper.

## 5  DETAILS OF LIKELIHOOD FUNCTION

We perform joint inference and learning based on the maximum likelihood principle [5]. Consider the likelihood function

$$p(l; \theta) = \frac{exp(-E(L \mid \theta))}{Z(\theta)}$$

Since in GTW the observation error is encoded in the edge cost, we assume a factorization of the likelihood function as follows:

$$p(L, (x, y) \mid \alpha, \beta, 2\sigma^2, \kappa) = p((x, y) \mid l; \alpha, \beta, 2\sigma^2) p(l; \kappa),$$

where the likelihood function for the data is

$$L_a(\alpha, \beta, 2\sigma^2) =: p((x, y) \mid l; \alpha, \beta, 2\sigma^2) = \frac{1}{z(\alpha, \beta, 2\sigma^2)} \exp\left(-\sum_{n=1}^{N} \sum_{(i,j) \in E_n} U_{n,(i,j)}(i, j)\right),$$

and where the likelihood function encoding neighborhood similarity is

$$L_b(\kappa) =: p(l; \kappa) = \frac{1}{z(\kappa)} \exp\left(-\sum_{(m,n) \in Neib} \sum_{i=1}^{|V_{n'}|} V(V_{n,i'}, V_{m,i'})\right).$$

Given labels $L$, we can estimate the parameters and hyperparameters, $\theta$, seeking to maximize the likelihood. Then we use the updated $\theta$ to get new labels, $L$. Multiplying these two likelihoods together, we can express the joint likelihood as a Gibbs distribution for the neighborhood-connected dual form graph based on extended DTW:

$$p(l; \theta) = \frac{1}{Z(\theta)} \exp\left(-\sum_{n=1}^{N} \sum_{(i,j) \in E_n} U_{n,(i,j)}(i, j) - \sum_{(m,n) \in Neib} \sum_{i=1}^{|V_n'|} V(V_{n,i}', V_{m,i}')\right).$$

Here $\theta := (\alpha, \beta, \sigma^2, \kappa)$ is the set of parameters and $E(L|\theta)$ is the dual form energy function, the exponent in $p(l; \theta)$. $V_{n,i}' \in V_n' \subset V_{gtw}$, which is a node coming from $G_n'$. $Z(\alpha, \beta, 2\sigma^2, \kappa)$ is the partition function, the proper normalization term. We will explain each parameter in the following.

The first double summation in the exponential is the single pair DTW cost, *i.e.*

$$\sum_{(i,j) \in E_n} U_{n,(i,j)}(i, j) := \text{cost}(P_n),$$

which is edge dependent. Given an edge $(i, j) \in E_n$, $U_{n,(i,j)}$ takes the matrix form

$$U_{n,(i,j)} = \begin{pmatrix} 0 & \dfrac{f_{n,dist}(i, j)}{2\sigma^2} + f_{direct}(i, j) + f_{off\_diag}(i, j) \\ \infty & 0 \end{pmatrix}.$$

Note that this similarity cost is not symmetric due to the asymmetric path constraints in the DTW problem. If $i = 0$ and $j = 1$, it is a cut and we have three terms. Assume nodes $i$ and $j$ match time points $k_1$ and $k_2$ along direction $d(i, j)$. It can take the value of east, north, or northeast, where northeast means the path in the primal graph is along the diagonal line. Then we have

$$f_{n,dist}(i, j) = (y_n(k_1) - x_n(k_2))^2.$$

The direction penalty is

$$f_{direct} = \begin{cases} 0, & \text{if } d(i, j) = northeast \\ \alpha, & \text{if } d(i, j) \in \{east, north\} \end{cases}.$$

In many cases, we prefer to penalize the path that is far away from the diagonal, so we have

$$f_{off\_diag} = \beta\sqrt{|k_2 - k_1|}.$$

The second double summation in $E(L|\theta)$ is the cost of neighboring pairs:

$$\sum_{i=1}^{|V_n'|} V(V_{n,i}', V_{m,i}') = cut(L_m, L_n), \quad (m,n) \in Neib.$$

We define the neighborhood similarity cost

$$V(i, j) = \begin{cases} 0, & if \ i = j \\ \kappa, & if \ i \neq j \end{cases}.$$

If neighboring nodes across different pairs have different labels, we incur cost $\kappa$.

# 6  PARAMETER AND HYPERPARAMETER ESTIMATION

## 6.1  Pseudo-likelihood based approach

First we estimate the smoothness term $\kappa$. Since $L_{b(\kappa)}$ involves the complicated partition function $z(\kappa)$, we replace it with the pseudo likelihood [5]:

$$L_b(\kappa) \approx \prod_{m:1 \leq m \leq N} \prod_{i=1}^{|V_m'|} \frac{\exp\left(-\sum_{n:(m,n)\in Neib} V(V_{m,i}', V_{n,i}')\right)}{\exp\left(-\sum_{n:(m,n)\in Neib} V(0, V_{n,i}')\right) + \exp\left(-\sum_{n:(m,n)\in Neib} V(1, V_{n,i}')\right)}.$$

As this pseudo-likelihood is concave in $\kappa$, we solve for it numerically by interior point methods.

We estimate other parameters in the primal domain. $\sigma^2$ is estimated by the residual error after applying the estimated warping function:

$$\hat{\sigma}^2 = \frac{1}{N_1 - 1} \sum_{n=1}^{N} \left\| y_n - \hat{W}_n(x_n) \right\|^2, \ N_1 = \sum_{n=1}^{N} \left| \hat{P}_n \right|.$$

The vector difference $y_n - \hat{W}_n(x_n)$ is computed by mapping all points in $x_n$ using the warping function and calculating the difference. We allow a single $x_{n,k}$ to be mapped to multiple $y_{n,k}$ estimates.

For $\alpha$, we use the pseudo-likelihood to approximate $p(P; \alpha)$ for the primal problem. All pairs are treated as independent. Within one pair, each step of the warping function is also treated as independent in the pseudo-likelihood:

$$p(P; \beta) \approx \prod_n p(P_n; \beta) \approx \prod_n \prod_t p(d_{n,t}; \alpha)$$

Here $d_{n,t}$ is the direction at the $t_{th}$ step (which is an edge in the primal graph) of warping function $P_n$. We define

$$p(d_{n,t}; \alpha) = \frac{\exp\left(-cost_\alpha(d_{n,t})\right)}{\sum_{i \in \{0,1\}} \exp\left(-cost_\alpha(i)\right)}$$

Here $cost_\alpha = f_{direct}$. We can then estimate $\alpha$ using maximum likelihood based on the warping function we estimated.

As an extension, we can estimate $\alpha$ as the direction consistency cost, or the smoothness of the path in terms of direction changing. For each step of the warping function, the probability is determined by whether the previous or next step has the same direction (diagonal moving or not).

$$p(P; \alpha) \approx \prod_n p(P_n; \alpha) \approx \prod_n \prod_t p(d_{n,t} | d_{n,t-1}, d_{n,t+1}; \alpha)$$

Here $d_{n,t}$ is the direction at the $t_{th}$ step for warping function $P_n$. Its value is 0 if it is a diagonal step (or edge) in the warping path and 1 if it is non-diagonal.

And we have:

$$p(d_{n,t} | d_{n,t-1}, d_{n,t+1}; \alpha) = \frac{p(d_{n,t}, d_{n,t-1}, d_{n,t+1}; \alpha)}{p(d_{n,t-1}, d_{n,t+1}; \alpha)}$$
$$= \frac{\exp(-cost_\alpha(d_{n,t}, d_{n,t-1}) - cost_\alpha(d_{n,t}, d_{n,t+1}))}{\sum_{i \in \{0,1\}} \exp(-cost_\alpha(i, d_{n,t-1}) - cost_\alpha(i, d_{n,t+1}))}$$

,

where $cost_\alpha$ is the cost of neighboring steps in the warping having the same or different directions, which is specified by $\alpha$.

The estimation of $\beta$ can be done using the same approach. Now the pseudo-likelihood becomes:

$$p(P; \beta) \approx \prod_n p(P_n; \beta) \approx \prod_n \prod_t p(e_{n,t}; \beta)$$

Here $e_{n,t}$ is the distance of path $n$ from the diagonal line at time $t$. And we define

$$p(e_{n,t}; \beta) = \frac{\exp\left(-cost_\beta(e_{n,t})\right)}{\sum_{i \in N} \exp\left(-cost_\beta(i)\right)}$$

Then we can estimate $\beta$ using the maximum likelihood. Here $cost_\beta = f_{off\_diag}$.

As the pseudo-likelihood only approximates the true likelihood function, it sometimes differs significantly from the true one for certain labelling patterns, which may cause the parameter estimation diverge. To deal with this, we add some perturbation to the labels by randomly flipping about 1% of them multiple times and then using the median to obtain the estimated parameters.

## 6.2  Cross validation based approach

We may use cross validation to estimate the hyperparameters. Below we discuss two possible ways of designing the cross validation scheme.

If we have multiple realizations of the time series pairs, we can hold out some of them as the validation set, on which the accuracy of warping paths estimated from the training set is evaluated. For each pair $(x_{val,n}, y_{val,n})$ in the validation set, given the corresponding warping paths (or functions) from the training set $P_n$, we can calculate the fitting error for pair $n$ in the validation set as $\left\| W_n(x_{val,n}) - y_{val,n} \right\|^2$, where $W_n$ is the warping function corresponding to path $P_n$. Repeating this for different choices of a hyperparameter $\kappa$, we can choose the best one.

Otherwise, we can divide each curve into two halves. For the first half of all curves, given a candidate choice for $\kappa$, we build the graph and estimate the parameters such as noise variance, as discussed above. Then we used the estimated parameters and the given hyperparameter to evaluate the overall cost function based on the collection of second half curves. We can scan the hyperparameter $\kappa$ over a range of interest in this way and choose the value of $\kappa$ that performs best. Then, given this choice of $\kappa$ fixed, GTW is run again, applied to the whole curves to learn the set of warping functions and all parameters.

**Supplement Figure 8 Synthetic calcium move. Left: movie at a certain time instant with Gaussian noise added. Right: all the FIUs generated and labelled by different colors.**

**Supplement Figure 9. Performance comparison of DTW and GTW on the synthetic data set with 24 FIUs under the same noise variance. The experiment on each FIU is repeated 10 times.**

# 7 EXPERIMENTS ON SYNTHETIC CALCIUM MOVIE

We generated synthetic calcium movies to quantitatively evaluate the performance of GTW for a realistic application, in addition to the qualitative comparison performed in the experiment part of the main text. In an astrocyte calcium movie, we define Function Independent Units (FIUs) as the smallest region that can contain a calcium wave activity [6]. A calcium wave activity consists of the rising, propagation and disappearing of a wave. Each activity is limited to an FIU and an FIU can only contain one activity at one time instant. Supplement Figure 8 (left) shows one time instant for synthetic data and the right part uses different colors to indicate different FIUs. 24 FIUs are simulated and totally there are 300 time points.

Assume we already know the exact locations of those FIUs. For each FIU, we use a ground truth curve for that FIU (without noise) as the common reference curve and the time series for all other pixels as test curves. The reference curve thus maps to all test curves. For both GTW and DTW, we set the off-diagonal cost to be 0.5. The noise variance of $0.05^2$ is assumed to be known. For each FIU we repeated the experiment 10 times. We calculated the ground truth warping functions using the noiseless data. For each FIU, we compared the performance of GTW and DTW, as shown in Supplement Figure 9. We can find that for all FIUs, GTW produces smaller error.

# 8 EXECUTION SPEED COMPARISON OF MAX FLOW ALGORITHMS

Max flow is used in the inference part of our model. Since we use pseudo-likelihood for parameter and hyperparameter estimation, the inference part takes the majority of the running time. We compare two popular max flow algorithms: Boykov-Kolmogorov (BK) [7] and iterative breadth first search (IBFS) [8] for our model. As shown in Supplementary Table 1, BK runs about three times faster. The only exception is the first iteration, where the similarity term is 0. This confirms that BK is suitable for generally sparse and grid-like structures. The simulation was performed with Matlab 2015b on a Dell T7910 workstation with dual 8-core Intel Xeon CPU E5-2630 @ 2.40GHz and 128 GB RAM. However, only a single process was used in the simulation. We use the C++ max flow implementation that can be found at [9].

**Supplement Table 1. Running time for max flow in seconds**

| Method | BK | IBFS |
|--------|-------|-------|
| Iter 1 | 0.339 | 0.338 |
| Iter 3 | 0.726 | 2.478 |
| Iter 5 | 0.698 | 2.595 |

The complexity of GTW depends on the window size, the number of time points, and the number of pairs. There are several possible solutions for large data sets: 1) reduce the dimension of the problem temporally or spatially or 2) break the problem into smaller pieces and solve them all in parallel. This strategy is widely used in fast graph cut implementations. Such an extension is possible for GTW and may be considered in future.