[Reviews · NeurIPS 2016]

Reviewer 1

Summary

This paper presents the problem of simultaneously aligning multiple pairs of related time series using dynamic time warping. The input to the problem consists of a set of pairs of time series, and a graph encoding a neighborhood structure on the pairs. The objective is to minimize the sum of the warping distances for each pair of time series and the sum of distances between the warping paths of neighboring pairs. The proposed solution starts from a min cut/max flow algorithm to solve standard DTW, and extends it to solve the simultaneous multiple alignment problem. An extension to learn DTW hyper-parameters using penalized likelihood-like criteria is also introduced. Results are provided for one simulated data set and one real data set.

Qualitative Assessment

Technical quality: The discussion of the proofs in the paper is very brief, making the technical sections quite dense. The supplemental material fills in the details and the proofs appear to be correct. One issue that could be addressed more clearly in the paper and the supplemental is the need for infinite weight reverse edges in the dual graph. This is currently not explained at all in the main paper other than stating these edges are important and later empirically demonstrating that not including them leads to violations of DTW warping constraints. The hyper-parameter tuning section was very brief and the algorithm is not clear. As the authors are using pseudolikelihood for some of the parameters the overall approach is certainly not proper maximum likelihood estimation. The supplemental material also describes additional ad hoc modifications introduced to help with convergence of the learning approach. This section of the paper could be cleaned up substantially. The empirical evaluation is one of the weaker aspects of the paper. The evaluation lacks depth as it relies on only one type of synthetic data set and one real data set. For the synthetic data set, a quantitative error evaluation is performed relative to independent DTW alignments in terms of comparing the estimated and true warping paths. This evaluation shows that the proposed method outperforms DTW in terms of mean error, but no standard errors or other statistics are reported to allow assessment of statistical significance of the results. The other experiment involves and application to cell imaging data. First, this application domain needs to be described as it may not be familiar to readers. Second, the analysis of this experiment is purely qualitative. It is clear that GTW shows increased spatial coherence relative to independent DTW, but beyond this, it is not clear that anything can be concluded from the results. Novelty/originality: The GTW problem dealt with in this paper has not been well studied in the literature. There is significant work on jointly aligning multiple curves, or alignment to a common reference, but this problem is distinct in that it concerns simultaneous alignment of multiple pairs of curves with warping functions that are allowed to be related in complex ways through the GTW graph. The use of max flow-based methods for DTW is not novel, but the authors build on it significantly to yield exact solutions to the proposed GTW problem, which is a novel contribution. Impact: This work could have a significant impact on application areas that make use of DTW. The ability to model interactions between warping paths using arbitrary graphs is quite powerful and allows for applications to spatiotemporal domains that could only be addressed using independent DTW previously. The authors might spend some more time in the paper describing more such application domains. Clarity and presentation: Another weakness of the paper is that the writing is not particularly clear in terms of grammar and the technical sections are quite dense. Some of the figure including 1(c) and 2(a) showing primal and dual graphs are quite hard to interpret at first.

Confidence in this Review

2-Confident (read it all; understood it all reasonably well)


Reviewer 2

Summary

This paper presents the problem of optimally aligning multiple time series. The work is motivated by the desire to leverage shared structure when warping/aligning multiple related signals to a single reference signal (or perhaps multiple reference signals). The authors develop “Graphical Time Warping” (GTW) to jointly model multiple time warping functions. The proposed objective function consists of two parts: 1) a term that penalizes the cost of each warping path (similar to standard DTW) and 2) a term that penalizes the dissimilarity between "adjacent" warping paths. Here, adjacent refers to time series that share an edge in the graph structure assumed by the user (either to time series share an edge or they don't). The authors then transform the optimization problem into a network flow problem and solve it using the max flow algorithm. Experiments on both real and simulated data are presented.

Qualitative Assessment

This work presents an efficient solution to the problem of aligning multiple signals when there is some assumed shared structured among the signals. Overall, the approach seems reasonable, but I have a number of questions for the authors: *In the problem setup, the task is described as jointly aligning N pairs of curves (x_n,y_n). However, the motivating examples appear to assume a single reference time series. In the case of multiple "reference signals", what assumptions are necessary regarding the underlying relationship governing the y's? *The dissimilarity function is defined as the area of the region bounded by the two paths. This relies on the boundary conditions being satisfied. Often, in the context of prefix or suffix matching, one relaxes the boundary conditions. Would the proposed methods no longer apply? What other dissimilarity functions are possible/did the authors consider? *In the problem setup, the authors assume a binary graph, in which similar nodes (i.e., time series) are either connected by an edge or not. Could the formulation be extended to a weighted graph? *A strength of the paper is that the authors consider the application of their proposed technique to both synthetic and real datasets. Comparisons between DTW and GTW, however, could be presented in a more rigorous manner, e.g., including error bars in Figure 3b. Have the authors compared DTW and GTW when used in the context of a classification (or clustering) problem? In the ML literature, DTW is often used to define a dissimilarity measure that can be used to classify time series. How much does one expect to gain when using GTW in place of DTW in such a scenario? *Are the authors familiar with the following work: “Mohammad Shokoohi-Yekta, Jun Wang and Eamonn Keogh (2015). On the Non-Trivial Generalization of Dynamic Time Warping to the Multi-Dimensional Case. SDM 2015.”? It seems particularly relevant. While it doesn’t make assumptions regarding any shared substructure among the signals, it does focus on the problem of multidimensional DTW. *The paper could benefit from a thorough proofread. There exist a number of typos/grammatical errors. e.g., "Euclidian" --> "Euclidean" (line 23)

Confidence in this Review

2-Confident (read it all; understood it all reasonably well)


Reviewer 3

Summary

In this paper, the authors propose a novel solution to the simultaneous alignment of multiple curves through the Graphical Time Warping algorithm, which preserves computational efficiency and global optimality from the original DTW procedure.

Qualitative Assessment

The paper deals with a very interesting problem by proposing a novel solution involving graph theory. Although no ingredient in the proposed pipeline is new, their combination is indeed clever, especially the formulation of the problem in a graph theoretical setup and the construction of the ad-hoc likelihood function. All proofs seem correct, and the section on the hyperparameter tuning is crucial for practical applications. The experimental section is convincing, although limited to two examples, a synthetic toy dataset and the astrocyte calcium image dataset. As a minor observation, a further example on a different real world dataset would have added value and strengthen the authors' claim. == post-rebuttal answer== I've read the rebuttal and confirm my review.

Confidence in this Review

3-Expert (read the paper in detail, know the area, quite certain of my opinion)


Reviewer 4

Summary

This paper proposed a interesting approach, called graphical time warping (GTW), to jointly align multiple sequence-pairs simultaneously. Traditional DTW only works in the case of aligning one single pair of time series, and in this paper, the author went one step further, developed GTW to align multiple pairs, formulated it as a max-flow problem, proved the theoretical global optimality of its solution to the objective function, and experimentally showed its superiority to DTW.

Qualitative Assessment

I like the algorithm GTW proposed in this paper. The authors formulate it as a max-flow problem and show the optimality of its solution. First, GTW is new to solve the multiple sequence-pairs alignment in a principled way; Second, re-formulating the Equation (3) as a max-flow problem is novel; Third, the paper is well written and easy to follow. What are needed to be improved: First, for the simulation experiments, the authors should compare with GTW WITHOUT diagonal constraints, i.e., in Fig(3b), the comparison should be among 4 algorithms: GTW, GTW+direct, DTW, DTW+direct. From the simulation, we know the alignment paths are approximately diagonal. But when reporting results, I would like to see what if the prior knowledge (diagonal alignment) is not enforced, how GTW performs. This result will show whether GTW works well under relaxed assumptions as well. Second, I am concerned about the practical applicability of GTW (in other words, how useful is GTW, could you find some practical applications, in which DTW does not work, but yours works? ). Although the authors used a case study to show GTW gets cleaner alignments than DTW, however, the cleaner alignment results are kind of under expectation, since graphical structures are explicitly enforced in GTW. Since there are no ground-truth alignments in your case study, the conclusion that GTW is better than DTW is subjective (to some extent). If you could find some practical applications with ground-truth alignment, then your conclusion is more convincing.

Confidence in this Review

2-Confident (read it all; understood it all reasonably well)


Reviewer 5

Summary

This paper studies the problem of aligning multiple time series when the similarity between the time series is given by a graph. They propose an objective function which adds a similarity cost term to the alignment cost. The key contribution of the authors is to show that the proposed problem can be solved by an instance of min-cut/max-flow problem. They also propose a maximum likelihood approach for tuning of the parameters.

Qualitative Assessment

I believe the problem proposed in the paper has some practical significance and the proposed algorithm is mathematically sound. The presentation of the ideas in the paper is also easy to follow. However, three aspects of the paper require clarification, in my opinion: Evaluation: It seems that the evaluation in the experiments section is not quite rigorous, and it is mostly qualitative (except Fig. 3(b)). This hinders the value of the paper. Scalability: Given the relatively small datasets in the experiments, the authors need to provide runtime analysis of the algorithm. The sentence in lines 100-101 increases my worries about scalability and practicality of the algorithm. Variable size time series: It seems that the distance metric between two warping paths only works if the time series have the same length. This limits practicality of the proposed approach. Also, while the notation table in the supplementary materials describes the notation in Section 4, without referring to it we cannot understand it.

Confidence in this Review

1-Less confident (might not have understood significant parts)